# Comparative Analysis of Complete Mitochondrial Genome of *Ariosoma meeki* (Jordan and Snider, 1900), Revealing Gene Rearrangement and the Phylogenetic Relationships of Anguilliformes

**DOI:** 10.3390/biology12030348

**Published:** 2023-02-22

**Authors:** Youkun Huang, Kehua Zhu, Yawei Yang, Liancheng Fang, Zhaowen Liu, Jia Ye, Caiyi Jia, Jianbin Chen, Hui Jiang

**Affiliations:** 1Anhui Provincial Key Laboratory for Quality and Safety of Agri-Products, School of Resource and Environment, Anhui Agricultural University, Hefei 230036, China; 2State Key Laboratory of Estuarine and Coastal Research, Institute of Eco-Chongming, East China Normal University, Shanghai 200241, China; 3School of Materials and Environmental Engineering, Chizhou University, Chizhou 247000, China; 4College of Life Sciences, Hainan Normal University, Haikou 571158, China

**Keywords:** *Ariosoma meeki*, Anguilliformes, mitochondrial genome, gene rearrangement, phylogenetic construction

## Abstract

**Simple Summary:**

In this study, we report the complete mitochondrial genome of *Ariosoma meeki* (Anguilliformes (Congridae)), the mitochondrial genome structure and composition were analyzed. We found the mitogenome of *A. meeki* has undergone gene rearrangement: The *ND6* and the conjoint *tRNA-Glu* genes were translocated to the location between the *tRNA-Thr* and *tRNA-Pro* genes, and a duplicated D-loop region was translocated to move upstream of the *ND6* gene. At the same time, we speculated the possible evolutionary process of gene rearrangement, and made the hypothesis of tandem repeat and random loss for explanation. The results of phylogeny also echo this inference in Anguilliformes.

**Abstract:**

The mitochondrial genome structure of a teleostean group is generally considered to be conservative. However, two types of gene arrangements have been identified in the mitogenomes of Anguilliformes. In this study, we report the complete mitochondrial genome of *Ariosoma meeki* (Anguilliformes (Congridae)). For this research, first, the mitochondrial genome structure and composition were analyzed. As opposed to the typical gene arrangement pattern in other Anguilliformes species, the mitogenome of *A. meeki* has undergone gene rearrangement. The *ND6* and the conjoint *tRNA-Glu* genes were translocated to the location between the *tRNA-Thr* and *tRNA-Pro* genes, and a duplicated D-loop region was translocated to move upstream of the *ND6* gene. Second, comparative genomic analysis was carried out between the mitogenomes of *A. meeki* and *Ariosoma shiroanago*. The gene arrangement between them was found to be highly consistent, against the published *A. meeki* mitogenomes. Third, we reproduced the possible evolutionary process of gene rearrangement in *Ariosoma* mitogenomes and attributed such an occurrence to tandem repeat and random loss events. Fourth, a phylogenetic analysis of Anguilliformes was conducted, and the clustering results supported the non-monophyly hypothesis regarding the Congridae. This study is expected to provide a new perspective on the *A. meeki* mitogenome and lay the foundation for the further exploration of gene rearrangement mechanisms.

## 1. Introduction

The gene rearrangement patterns among vertebrate mitogenomes were initially considered to be conservative. The convincing nature of this concept was reinforced after the complete mitochondrial genome of mammalians was uncovered and was proven to share the same genetic order [1]; it was then weakened as more gene-rearranged species have been reported. The improvement of sequencing technology has led to an increase in the discovery of gene rearrangement in vertebrate mitogenomes over the past few years [2,3]. Approximately 4% of the reported fish mitogenomes revealed gene rearrangements [4], such as transfer, translocation, and inversion, involving 34 families, according to an analysis covering 1300 species from the National Biotechnology Information Center (NCBI) database. The occurrence of gene rearrangement has been considered as a valuable evolutionary trace of retrospective species genesis in previous research [5,6]. The Anguilliformes, which are distributed in a wide variety of environments, include 2 suborders, 19 families, and 147 genera, and comprise approximately 600 species in total [7]. To date, two types of gene arrangement patterns (traditional gene arrangement and special gene rearrangement) have been reported; the latter occurs in a small percentage of the Anguilliformes population [8,9].

The conjectures regarding gene rearrangement in mitochondrial genomes that have been proposed so far can be classified into three main hypotheses [10]. In the first scenario, Poulton et al. [11] initially analyzed human mitochondrial rearrangements and found that mitochondrial genomes were involved in DNA strand-breaking and reconnection, before homologous recombination. This model of gene rearrangement has been shown in subsequent studies involving mussels, birds, and frogs, being used as a practical tool to assist interpretation and analysis [12,13]. The second model was proposed by Arndt et al. [14] and has been widely accepted; it features the interpretation of tandem replication and the random loss of mitochondrial genomes (TDRL model). In this model, the mitochondrial rearrangement was considered to be the result of structural variation caused by the random deletion of duplicates after tandem replication in some genes [15]. This hypothesis has been adopted in many gene rearrangement studies [16,17]. The third model was proposed by Lavrov et al. [18] in the mitochondrial genome report of millipedes. The model indicates that random loss is activated after full replication by mitochondrial genes, and the loss ultimately depends on the polarity and position of the genes. Unlike the TDRL model, this one emphasizes tandem repetition and nonrandom loss (TDNL model). Given the above pioneering and characteristic hypotheses, there is no conclusive interpretation of how gene rearrangements occur and what their implications are. Slowly but surely, details about facilitation in gene rearrangement are starting to trickle in with the enrichment of the genomic databases.

Comparative genomics analysis is widely regarded as an efficient approach for phylogenetic analysis, owing to the richness of molecular information that is characterized by maternal inheritance, a high rate of evolution, and a relatively low rate of intermolecular recombination [19]. Generally, most eels are snake-like in size, featuring narrow gill pores and no pelvic fins [20]. For these reasons, much fuzziness is involved in their identification and the corresponding phylogenetic research. In this context, the advantages of molecular approaches become prominent when it comes to species with insufficient morphological data. Nevertheless, genetic information for most Anguilliformes species is insufficiently disclosed, and the location of each species within the Anguilliformes phylogeny remains elusive [8], resulting in poor understanding and controversies.

In this paper, we report a new version of the complete mitochondrial genome sequence of *A. meeki*; the genetic composition and arrangement of *A. meeki* have been described in detail and a comparative genomic analysis has been conducted for *A. meeki* (this study) and *Ariosoma shiroanago*. Interestingly, gene rearrangement was detected in both *Ariosoma* mitogenomes, differing from the genetic features of two published *A. meeki* mitogenomes (KX641476 and MN616974), which have been marked as “Unverified” in the NCBI database. Therefore, we have performed the correct sequence test and publication. At the same time, for this study, we have carried out a systematic analysis of the species evolution of Anguilliformes.

## 2. Materials and Methods

### 2.1. Sample Collection, DNA Extraction, and PCR Amplification and Sequencing

Individual *A. meeki* samples were collected using traditional trawling methods that have been approved in Zhoushan City, Zhejiang Province, China (30°40′30″ N, 121°20′28″ E). After the samples were collected, we preserved them with 95% ethanol and kept them permanently in a freezer at −80 °C in the museum of the National Marine Facility’s aquaculture engineering technology research center. Total DNA was extracted from the tissue of 6 individual specimens’ muscles, using an Aidlab Genomic DNA Extraction Kit (Beijing, China), following the manufacturer’s instructions. The complete mitochondrial genome of *A. meeki* was sequenced by Sangon Biotech (Shanghai, China), and the design primers were based on the published mitochondrial genome and then amplified in strict accordance with the requirements of the kit (Takara, China) [21].

### 2.2. Sequence Analysis and Assembly and Mitochondrial Genome Annotation

The complete mitogenome of the invertebrate genetic sequence was annotated using the MITOS web server (http://mitos2.bioinf.uni-leipzig.de/index.py, (accessed on 15 July 2022) [22,23] and was then manually corrected. After correcting the sequenced DNA fragment results, the CodonCode Aligner 5.1.5 (CodonCode Corporation, Dedham, MA, USA) was applied to splice the fragments and complete the creation of the mitotic genome. The Sequin software (version 15.10) was used to mark and annotate the complete mitochondrion genome sequence, to determine the location and boundary of protein-coding and the ribosomal RNA gene; the results were verified via NCBI-BLAST (http://blast.ncbi.nlm.nih.gov, accessed on 4 July 2022) comparison. MitoFish (version 3.69, http://mitofish.aori.u-tokyo.ac.jp/annotation/input.html, (accessed on 4 July 2022) was used to predict the sequence characteristics of the *A. meeki* mitochondrial annular genome and to map the cyclical gene. The gene rearrangements were defined using the CREx program (http://pacosy.informatik.uni-leipzig.de/crex, (accessed on 28 July 2022) [24]. The relative synonymous codon usage (RSCU) values were analyzed with MEGA 11.09 [25]. Composition skew values were calculated according to the following formulas: AT-skew = (A − T)/(A + T); GC-skew = (G − C)/(G + C) [26].

### 2.3. Phylogenetic Analyses

Thirty-one complete Anguilliformes mitochondrial genomes were downloaded from GenBank (https://www.ncbi.nlm.nih.gov/genbank/, (accessed on 28 July 2022) for phylogenetic studies (Table 1). Saccopharyngiformes have been thought to be closely related to Anguilliformes [27]. Therefore, two Saccopharyngiformes species, *Eurypharynx pelecanoides* and *Saccopharynx lavenbergi*, were selected as the outgroup. In the analysis, 12 PCGs sequences were selected in order to construct a phylogenetic tree using DAMBE, version 7.2.3 [28]. These PCGs did not include *ND6* since the base composition was more irregular than other sequences and led to poor phylogenetic performance [29]. Sequences were aligned with default parameters, using Clustal X 2.0 [30], and were manually checked using BioEdit [31]. Ambiguous sequences were eliminated using Gblock [32]. Substitution vs. the Tamura–Nei (TN93) genetic distance in pairwise comparisons was used to test for substitution saturation, using DAMBE (version5.3.19) [33]. The third codon positions showed significant saturation, which, as a result, were defined only as purines and pyrimidines (3RY) [34]. The phylogenetic analyses were conducted utilizing the MrBayes 3.2.6 and PhyML80 software, based on Bayesian inference (BI) and maximum likelihood (ML), respectively [35,36]. The best-fit models of nucleotide substitution for each of the sequences were selected, using MrModelTest 2.2, from 33 models [37]. ML analysis uses bootstrap analysis (1000 repetitions) to verify the relative support levels [38]. Bayesian analysis was performed using default settings over four independent sets; the average standard deviation of split frequencies was < 0.01, the estimated sample size was >200, the potential scale reduction factor approached 1.0, and all parameters were checked with Tracer v. 1.6 [39]. The resulting phylogenetic trees were visualized using FigTree v. 1.4.4 and its tool (https://itol.embl.de, (accessed on 29 July 2022) [40].

## 3. Results and Discussion

### 3.1. Genome Structure and Composition

Compared to the published mitochondrial genome of bony fish, the complete mitogenome of *A. meeki* (17,695 bp) is within the normative range (16,099–18,247 bp) (Table 1). In all, 37 typical coding regions (13 PCGs, 22 tRNAs, 2 rRNAs) and 1 origin of replication between the WANCY structures (*tRNA-Trp*, *tRNA-Ala*, *tRNA-Asn*, *tRNA-Cys*, and *tRNA-Tyr*) were found in the mitogenome of *A. meeki*. However, two control regions (D-loops) were detected next to the *tRNA-Thr* and *tRNA-Pro* genes, respectively; the features of duplicated D-loops were distinct from most vertebrate mitogenomes [1]. The gene order of the *A. meeki* mitogenome was also distinctive. It indicated that the *ND6* and *tRNA-Glu* genes were translocated upstream of *tRNA-Pro* and downstream of *tRNA-Thr*, respectively. The additional D-loop was translocated to the site ahead of *ND6* as well; such structural heterogeneity is regarded as an uncommon trait among vertebrate mitogenomes [6,44].

The mitogenomic compositions and corresponding gene order for most bony fish are considered to be relatively conserved. Previous research revealed that the mitochondrial gene structure of homologous species could be analogous. For instance, many Anguilliformes species, such as the *Anguilla*, *Simenchelys*, and *Synaphobranchus* populations, revealed typical mitogenomic compositions (37 coding regions and two non-coding regions), and the gene arrangement among them showed a high degree of consistency, while it would be another story when it comes to other Anguilliformes species, such as *Facciolella*, *Ariosoma*, and *Muraenesox*, of which the species in the *Ariosoma* genus revealed a distinct gene rearrangement pattern [8]. In the mitogenome of *A. meeki*, an additional control region was detected, while one coding protein, *ND6*, underwent translocation (Figure 1). Compared with other non-rearranged species, such a genetic phenomenon between *Cytb* and *ND6* can be found across the published *Ariosoma* mitogenome (GenBank accession number: AP010861). Nevertheless, there is only one control region in the reported mitogenome of *A. shiroanago.* In this study, we performed a comparison and analysis of other genetic locations between *A. meeki* and *A. shiroanago* (Table 2 and Table 3) (Figure 2, Figure 3 and Figure 4). The overall base composition was 28.93% (A), 25.39% (C), 26.14% (T), and 19.54% (G) for *A. meeki*, and the overall base composition was 32.54% (A), 23.90% (C), 26.85% (T), and 16.71% (G) for *A. shiroanago* (Table 2 and Table 3). Typically, most of the molecular components in mitochondrial genomes were closely linked with each other; 11 intergenic spacers consisting of 107 base pairs were observed in the *A. meeki* mitogenome. The largest region contained 49 pairs of nucleotides and was located between *ND5* and *Cytb*; the smallest area was formed by a single 1 bp between *Cytb* and *tRNA-Thr.* As for the *A. shiroanago* mitogenome, the corresponding length thresholds for non-coding regions were 154 bp and 41 bp, which were situated between *ND6* and *tRNA-Thr, Cytb,* and *ND5,* respectively.

### 3.2. PCGs and Codon Usage

Thirteen PCGs of the *A. meeki* and *A. shiroanago* mitogenomes contained 11,477 bp and 11,482 bp encoding 3825 and 3827 amino acids, respectively. The PCGs consisted of seven NADH dehydrogenases (*ND1, ND2, ND3, ND4, ND4L, ND5,* and *ND6*), three cytochrome b oxidases (*COX1, COX2,* and *COX3*), two ATPases (*ATP6* and *ATP8*), and one cytochrome b (*Cytb*) [31]. Twelve genes were encoded in the H-strand, with only *ND6* genes in the L-strand. Gene arrangement in both mitogenomes was similar to that of the typical vertebrate mitogenome [45,46].

The initiation codons for most PCGs were ATG, except for the *COX1* gene, which was initiated with GTG (Table 2). The utilization of such an extraordinary initiation codon for the *COX1* gene is observable in most teleostean mitogenomes [47]. In the *A. meeki* mitogenome, five PCGs (*ATP8*, *ATP6*, *COX3*, *ND4L*, and *Cybt*) ended by TAA, while 5 PCGs (*ND1*, *ND2*, *ND3*, *ND5*, and *ND6*) were terminated by TAG, *COX2* and *ND4* terminated with a single T, and *COX1* was stopped by AGA. The phenomenon of the coding proteins ended by incomplete codons is widespread in invertebrate as well as vertebrate mitogenomes [48,49]. The probable mechanism adopted to explain this occurrence is that the codon TAA has generated the subsequent polyadenylation process through transcription [50]. Three PCGs in the *A. shiroanago* mitogenome owned different types of stop codons, which showed *ND5* and *ND6* terminated with TAA and *COX1* terminated with AGG.

The modes for the codon usage of 13 PCGs between the *A. meeki* and *A. shiroanago* mitogenomes are shown in Figure 2. The amino acids primarily used in PCGs for the previous species could be Leu1, Ser2, Pro, and Thr, with Leu1, Leu2, Ser1, and Ser2 for the latter. The relative synonymous codon usage (RSCU) analysis in two *Ariosoma* species, in terms of the third position, is depicted in Figure 2. The utilization among two- and four-fold degenerate codons presented an overall bias toward those codons that are abundant in A.

### 3.3. Transfer RNAs, Ribosomal RNAs, and D-Loops

Twenty-two *tRNAs* in *A. meeki* and *A. shiroanago* were detected, based on their unique codons. Fourteen *tRNAs* were located on the heavy chain, and the rest were *tRNAs* (*tRNA-Gln*, *tRNA-Ala, tRNA-Asn, tRNA-Cys, tRNA-Tyr, tRNA-Ser, tRNA-Glu,* and *tRNA-Pro*), which were found on the light chain (Figure 1). A typical clover structure was detected among most *tRNAs* in both species, with the exception of *tRNA-Ser1* (Figure 3 and Figure 4), which was unable to form a stable clover structure due to the lack of a complete dihydrouridine arm [51]. Twenty-two *tRNAs* contained 1560 base pairs in the *A. meeki* mitogenome, with 1562 in the *A. shiroanago* mitogenome. The length of each tRNA gene ranged from 64 to 76 bp in the mitogenome of *A. meeki* and from 66 to 76 bp in the mitogenome of *A. shiroanago*. The base composition was 29.03% (A), 27.12% (T), 20.45% (C), and 23.40% (G) for the former, and 29.71% (A), 28.17% (T), 20.23% (C), and 21.90% (G) for the latter. The anticodons of the mitogenomes in two *Ariosoma* species reflected the same usage pattern (Table 2).

The genes *12S* and *16S* were found to be surrounded by *tRNA-Phe* and *tRNA-Leu1* and were separated by *tRNA-Val* in the *A. meeki* and *A. shiroanago* mitogenomes (Table 3). The total length of *rRNAs* was 2652 bp and 2687 bp in both *Ariosoma* mitogenomes, respectively. The base compositions of the *rRNAs* were 33.78% (A), 20.06% (T), 22.25% (C), and 23.91% (G) for the mitogenome of *A. meeki*, and 36.55% (A), 20.54% (T), 22.29% (C), and 20.62% (G) for the mitogenome of *A. meek*. The AT-skew and GC-skew values were 0.04 and 0.26 for the former and 0.28 and −0.04 for the latter. Such a base bias revealed the higher percentages of the adenine and cytosine nucleotides of two *rRNA* genes in both mitogenomes.

In the *A. meeki* mitogenome, two D-loops were detected downstream of *tRNA-Thr* and *tRNA-Pro*, respectively. The total length of the D-loops was 1929 bp (969 bp for D-loop1 and 960 bp for D-loop2), with an identical AT bias of 62.00% (Table 2 and Table 3). Compared to other components, both D-loops presented a higher percentage in terms of AT bias. For this reason, the D-loop region, as the main non-coding area, has been called the “AT-rich region”. The AT- and GC-skew values were 0.17 and −0.17 in both D-loops, reflecting the abundance of adenine and cytosine and the thymine deficiency and guanine. The termination of heavy chain replication could be located in two D-loops, based on the palindrome element motifs, such as “TACAT” and “ATGTA” [52] (Figure 5).

The exploration of a substitution saturation index was implemented, utilizing the aligned PCGs of 33 Anguilliformes mitogenomes. Compared to the third codon, substitution in the first and second codons was relatively low (Figure 6). The impact on the whole amino acid released by the mutation of the third codons is rather vulnerable, in comparison with that of the first two codons, which possess a constrained evolving freedom [53,54]. The substitution in the third codons, even in insect mitogenomes, remains observable, as the amino acid products were relatively insensitive to the variation in the third codons [55].

### 3.4. Gene Rearrangement

Since one of the published *A. meeki* mitogenomes (MN616974) was not annotated, another *A. meeki* mitogenome (KX641476) was selected to conduct a comparative genomic analysis. The results indicated that there were not any other genes behind the *tRNA-Thr* in KX641476; three typical genes, *ND6, tRNA-Glu,* and *tRNA-Pro*, cannot be identified, as in most mitochondrial genomes of vertebrates. On the contrary, both *A. meeki* (this study) and *A. shiroanago* have experienced gene rearrangement, with the rearranged protein-coding *ND6*, transfer RNA *Glu,* and a duplicated non-coding region, D-loop (Figure 7A).

Generally, a teleostean group possesses unique or highly similar gene arrangements. Nevertheless, two types of gene arrangements have been detected in the mitogenomes of Anguilliformes. As observed in the mitogenomes of *Anguilla marmorata* (Anguillidae), *Ilyophis brunneus* (Synaphobranchidae), *Serrivomer sector* (Serrivomeridae), and *Gymnothorax minor* (Muraenidae), which possess the typical mitochondrial structure, the genes *ND6* and the conjoint *tRNA-Glu* were found to be typically situated between *ND5* and *Cytb*, among which the *Cytb* was sandwiched by *tRNA-Glu* and *tRNA-Pro,* without the insertion of a D-loop between *Cytb* and *ND6* [56] (Figure 7C). In the mitogenomes of *A. meeki* and *A. shiroanago, ND6* and *tRNA-Glu* were translocated upstream of *tRNA-Pro* and downstream of *tRNA-Thr*, respectively, accompanied by the insertion of a larger non-coding region. Nucleotide composition and the base arrangement of two control regions display similarities to a high and low degree in the mitogenomes of *A. meeki* and *A. shiroanago*, respectively.

Three typical models are expected to explain the gene rearrangement features in *Ariosoma* mitogenomes. The model, based on the recombination hypothesis, was initially proposed for gene rearrangement in the nuclear genome and was generally adopted to explain small-fragment exchanges and inversion events in mitogenomes. However, our comparative analysis shows that the *Ariosoma* mitogenomes have undergone gene translocation and genome-scale expansion. Therefore, the discovery of mitochondrial gene rearrangement in *Ariosoma* mitogenomes is too far-fetched to be explained by this model. The TDNL model placed an emphasis on non-random loss. In this case, the duplication followed by the loss of genes is a predesigned resorting to the corresponding transcriptional polarity and position in the genome; the genes are clustered in the same polarity (light- or heavy-strand coding) and the gene order remains unchanged (for example, the GCT cannot be produced by gene loss alone during the duplication from TCG to TCGTCG) [18,57]. This hypothesis does not apply to the findings, as the structural variation in *Ariosoma* mitogenomes is not the result of alterations in the genes’ transcriptional polarity. The TDRL model underlined the rearrangement, based on the incomplete deletion of repeated genes [58]. In the *A. meeki* mitogenome, an extra D-loop was found downstream of *tRNA-Thr*; both D-loops revealed a high degree of resemblance in terms of nucleotide composition and bas-e arrangement. In addition, *ND6,* combined with the *tRNA-Glu* genes, was translocated from the stream downstream of *ND5* to the upstream of *tRNA-Pro*. In the TDRL model, the intergenic spacers or pseudogenes were commonly detected in the rearrangement region [59,60]. In the mitochondrial genome of *A. meeki*, the existence of a considerable interval among *ND5* and *Cytb* further indicated the bias to this model (Table 2). In light of that, the TDRL model was suggested for the interpretation of rearrangement events in the mitochondrial genomes of *A. meeki and A. shiroanago*; among them, more detections of intergenic spacers or pseudogenes may occur in the *A. shiroanago* mitogenome during the random loss process.

Using the hypothesis proposed that is based on the TDRL model, we reproduced the ins and outs of the gene rearrangement in the *A. meeki* mitogenome (Figure 7B). The probable progressive process can be illustrated as follows: firstly, the duplication of the typical mitochondrial gene block (“*ND6*”-“*tRNA-Glu*”-“*Cytb*”-“*tRNA-Thr*”-“*tRNA-Pro*”-“D-loop”) commenced and led to the consecutive intermediate product (“*ND6*”-“*tRNA-Glu*”-“*Cytb*”-“*tRNA-Thr*”-“*tRNA-Pro*”-“*D-loop*”-“*ND6*”-“*tRNA-Glu*”-“*Cytb*”-“*tRNA-Thr*”-“*tRNA-Pro*”-“D-loop”); secondly, the randomly lost events took place in the mitochondrial components, while the redundant genes (*ND6, tRNA-Glu, Cytb, tRNA-Thr,* and *tRNA-Pro*) were randomly lost during that process; finally, the gene-rearranged structure was formed as (“*ND5*”-“*Cytb*”-“*tRNA-Thr*”-“D-loop”-“*ND6*”-“*tRNA-Glu*”-“*tRNA-Pro*”-“D-loop”), although there might be an explanation as to why there is a strong resemblance between the two D-loops, in terms of nucleotide composition [44].

### 3.5. Phylogenetic Analysis

Fourteen highly related families (*Eurypharynx pelecanoides* and *Saccopharynx lavenbergi*, functioning as outgroups) were selected to explore the phylogenetic location of *A. meeki*. The phylogenetic analysis was implemented with the information revealed by phylogenetic trees (BI and ML), based on 12PCGs (the highly varied sections were manually corrected). The topology of both trees presented a high degree of consistency (Figure 8) and indicated that families (e.g., the Nettastomatidae, Muraenesocidae, and Colocongridae) with gene rearrangement grouped together and formed an independent clade, while the rest of the families (e.g., the Anguillidae, Synaphobranchidae, and Muraenidae) with the traditional gene order formed another clade. The result implied the evolution and origin of the diversified Anguilliformes species, as this pattern of the presence/absence of gene arrangement may be a potential phylogenetic marker for the identification of this morphologically conservative group of eels at the suborder level.

Both the BI and ML phylogenetic trees revealed that *A. meeki* and *A. shiroanago* were strongly related to each other; they formed a separate clade of Congridae (BI posterior probabilities [PP] = 1; ML bootstrap [BP] = 100). Consistent with previous studies, Congridae was closely related to the Muraenidae, Myrocongridae, and Synaphobranchidae [8]. Except for the family Congridae, which was deemed to be the non-monophyletic taxa [10], the rest of the families unanimously revealed the monophyletic clade, while two phylogenetic trees highly supported the non-monophyletic Congridae, and two clades were observed; such clustering results were compatible with those of previously published studies [61]. Regarding the clustering results released by an increasing number of phylogenetic studies on Anguilliformes, an independent subfamily (including *Ariosoma*) may be forming or may even have formed. This may offer a new perspective on the Anguilliformes taxonomy, but more available mitogenomes are needed to test this hypothesis.

The phylogeny within Anguilliformes can be further routed into two major clades: one consists of eels that presented non-gene rearrangement patterns (Synaphobranchidae, Nemichthyidae, and Serrivomeridae), while another one comprised eels demonstrating gene rearrangement features (Nettastomatidae, Congridae, Ophichthidae, Muraenesocidae, and Derichthyidae). Similar consequences can be identified in other phylogenetic research on Anguilliformes [10]. The phylogenetic tree implied the evolutionary relationship of diversified Anguilliformes species and revealed the non-monophyly of the Congridae. According to previous studies, gene rearrangements may have originated from a common ancestral species for the Congroidei population. These features, which are found on the mitochondrial genomes of the partial population, were viewed as cryptic evolutionary population traces to distinguish them from other related species that possess a relatively conserved morphology [20,62]. However, with few available Anguilliformes species with gene rearrangements, a thorough confirmation of this hypothesis or clarification of the phylogenetic relationships among Anguilliformes species could be unattainable; depictions of the phylogenetic relationships resort to abundant genomic resources among this taxon would be more authoritative.

Given the remarkable resemblance in terms of the composition (Table 2 and Table 3) and the codon usage (Table 2) (Figure 2) of PCGs, the structure of tRNA (Figure 3 and Figure 4) and the results of phylogenetic analysis (Figure 8) between *A. meeki* and *A. shiroanago*, it may be a proof of a congener and two species that belong to the Congridae genetically. The variations in mitogenome length may be attributed to incomplete sequencing or annotation. One of the D-loops that should have been located between *tRNA-Thr* and *ND6* was found to be missing.

## 4. Conclusions

The arrangement of the mitochondrial genome is consistent in most vertebrate mitogenomes; two kinds of gene arrangements have been revealed in the order Anguilliformes, which leads us to some confusion about the mechanism involved. In particular, some Anguilliformes populations were known to converge into non-monophyletic results, such as species in the Congridae and Nettastomatidae [9]. As of now, there is no comprehensive elaboration for this, since the relevant research is in its infancy [63].

In this study, we reported the complete mitochondrial genome of *A. meeki*, analyzed the corresponding genomic information, compared it with the reported mitogenomes of *A. meeki* and congeneric species, and depicted the phylogenetic relationship among Anguilliformes species. The complete mitogenome of *A. meeki* exhibited a rearrangement of gene order, in contrast to the typical vertebrate mitogenomes; *ND6* and *tRNA-Glu* genes were translocated upstream of *tRNA-Pro* and downstream of *tRNA-Thr*, respectively, accompanied by the insertion of a D-loop. Both D-loops exhibited a high degree of resemblance in terms of nucleotide composition and base arrangement. The phylogenetic analysis indicated that *A. meeki* was closely related to its congeneric, *A. shiroanago*. The two species clustered together as a separate clade of the Congridae family. According to previous phylogenetic studies on Anguilliformes, the non-monophyly of Congridae was reflected by utilizing a comparative genomic analysis, based on 33 mitochondrial genomes of the Anguilliformes species. Both phylogenetic trees strongly support the non-monophyly of Congridae, offering a theoretical basis for more advanced phylogenetic studies of Anguilliformes. In addition, our results provide new insight into the evolution of *A. meeki* and an in-depth understanding of the mechanism of gene rearrangement in the *A. meeki* mitogenome.

## Figures and Tables

**Figure 1 biology-12-00348-f001:**
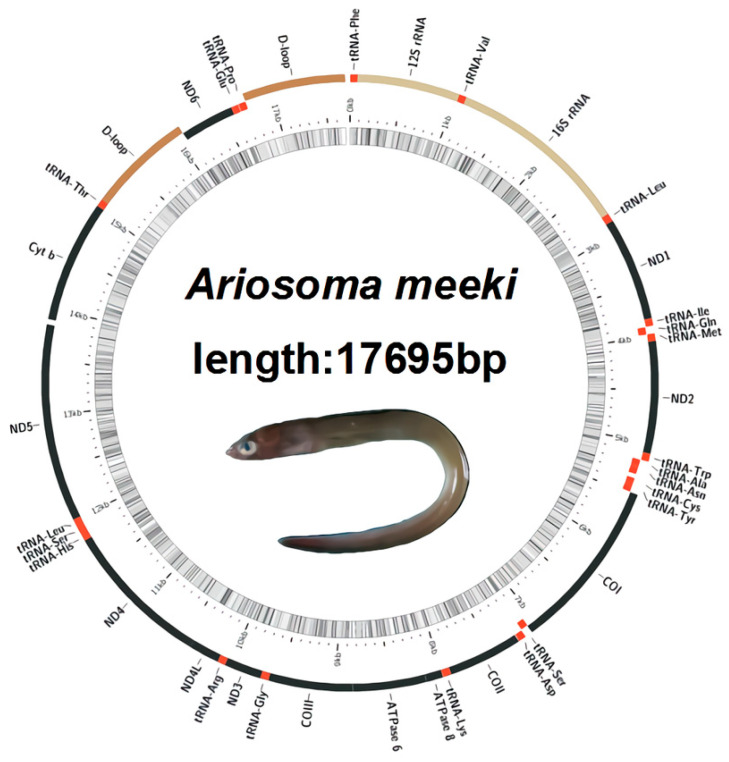
Gene map of the *Ariosoma meeki* mitogenome.

**Figure 2 biology-12-00348-f002:**
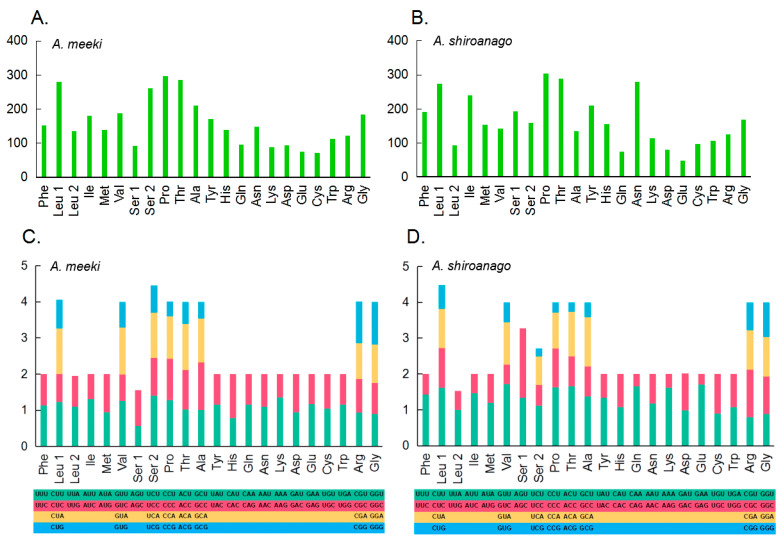
Amino acid composition in the mitogenome of *A. meeki* (**A**) and *A. shiroanago* (**B**) (The *x-* and *y*-axis refer to the amino acid composition and the number of each amino acid in 13 PCGs, respectively). Relative synonymous codon usage (RSCU) in the mitogenome of *A. meeki* (**C**) and *A. shiroanago* (**D**). (The *y*-axis represents the usage frequency of the corresponding amino acid codons in 13 PCGs. Different colors represent the different codons in the amino acids).

**Figure 3 biology-12-00348-f003:**
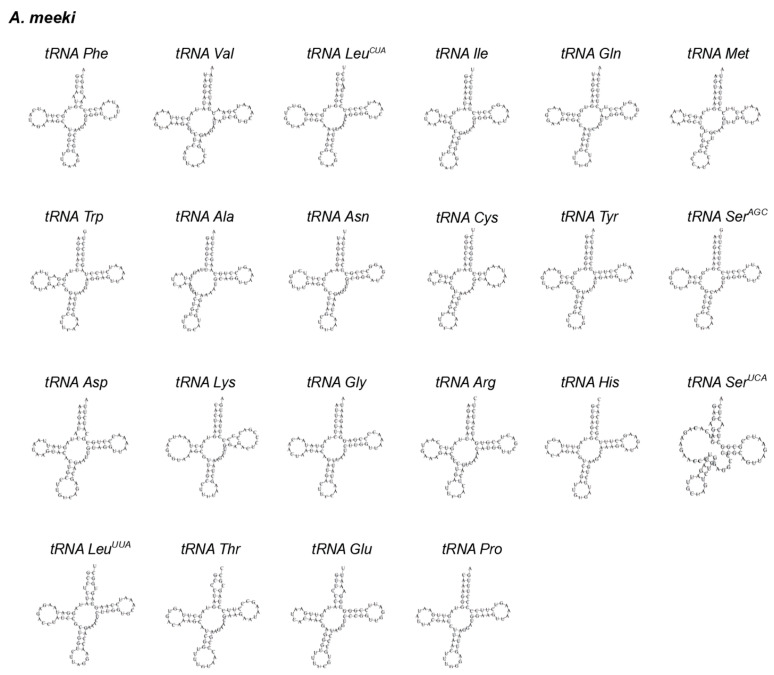
The cloverleaf structure of 22 *tRNAs* in the mitochondrial genome of *A. meeki*.

**Figure 4 biology-12-00348-f004:**
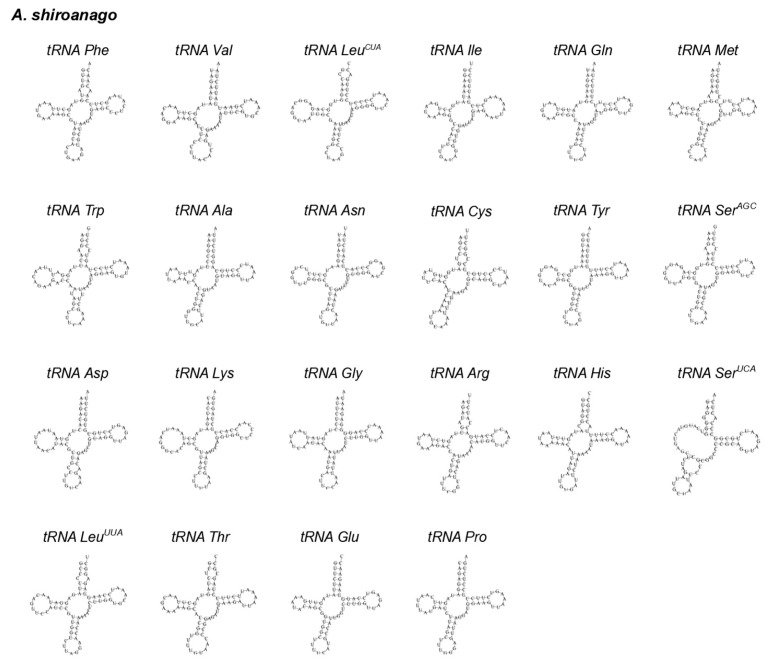
The cloverleaf structure of 22 *tRNAs* in the mitochondrial genome of *A. shiroanago*.

**Figure 5 biology-12-00348-f005:**
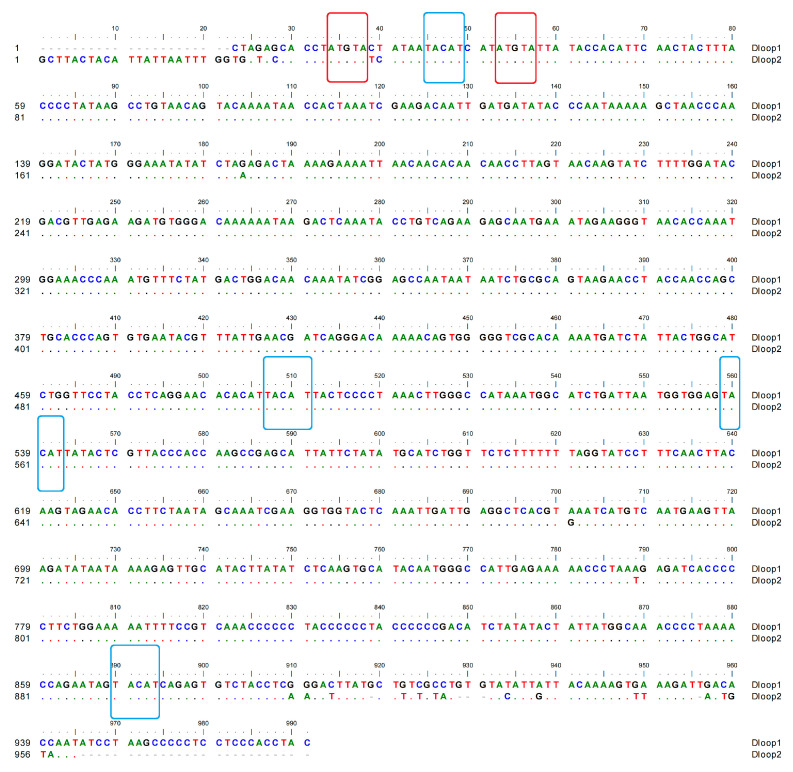
Compositional features of the control region of the *A. meeki* mitochondrial genome. The palindromic motif sequences “TACAT” and “ATGTA” are marked in blue and red, respectively.

**Figure 6 biology-12-00348-f006:**
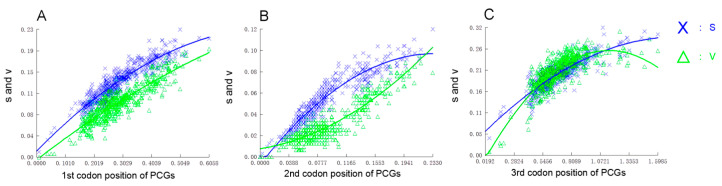
Nucleotide substitution saturation plots for all 13 PCGs. (**A**) First codon positions; (**B**) second codon positions; (**C**) third codon positions. Plots in blue and green indicate the transition and transversion, respectively.

**Figure 7 biology-12-00348-f007:**
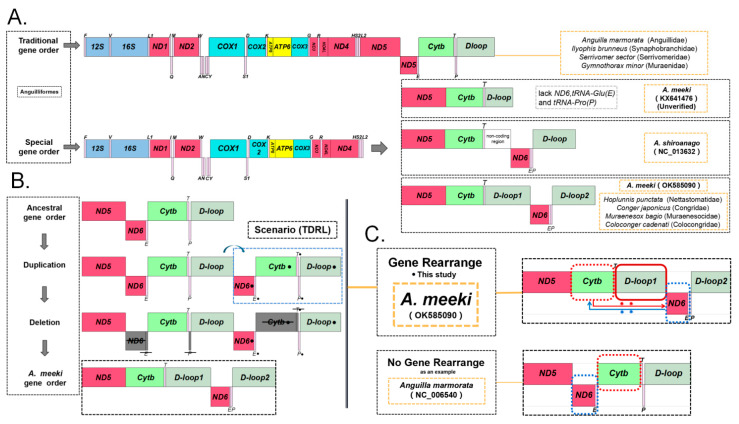
Analysis of the *A. meeki* mitochondrial gene rearrangement. (**A**) Structure of the mitochondrial genome and gene rearrangement of *A. meeki;* (**B**) replication and random deletion during gene rearrangement; (**C**) structural comparison of the gene rearrangement and non-gene rearrangement.

**Figure 8 biology-12-00348-f008:**
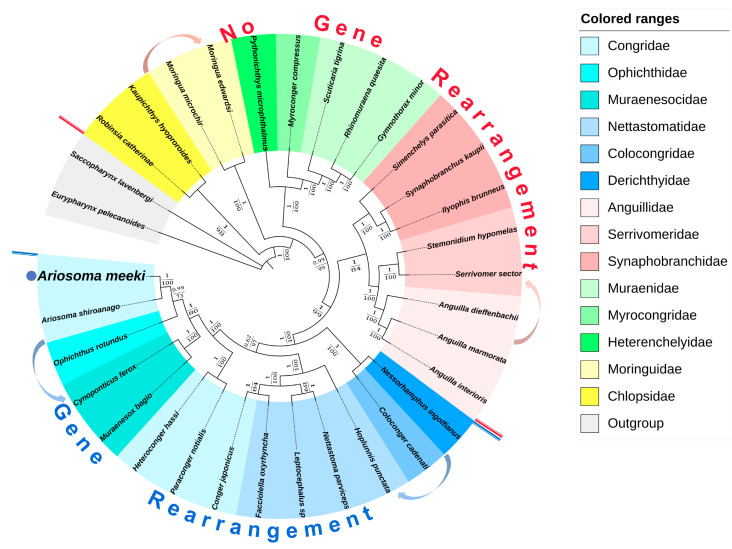
Phylogenetic tree of the Anguilliformes inferred from the nucleotide sequences of 12 PCGs (except for *ND6*) using the Bayesian inference (BI) and maximum likelihood (ML) methods. The numbers on the branches indicate posterior probability (BI) and bootstrap (ML).

**Table 1 biology-12-00348-t001:** List of the 31 Anguilliformes species and 2 outgroups used in this paper.

Species	Family	Size (bp)	Accession No.	References
*Leptocephalus* sp.	Nettastomatidae	18,037 bp	NC_013615	[8]
*Nettastoma parviceps*	Nettastomatidae	17,714 bp	NC_013625	[8]
*Facciolella oxyrhyncha*	Nettastomatidae	17,789 bp	NC_013621	[8]
*Hoplunnis punctata*	Nettastomatidae	17,828 bp	NC_013623	[8]
*Paraconger notialis*	Congridae	17,729 bp	NC_013630	[8]
*Heteroconger hassi*	Congridae	17,768 bp	NC_013629	[8]
*Conger japonicu*	Congridae	17,778 bp	KR131863	[8]
*Ariosoma shiroanago*	Congridae	16,922 bp	NC_013632	[8]
*Ariosoma meeki*	Congridae	17,659 bp	OK585090	This Study
*Cynoponticus ferox*	Muraenesocidae	17,822 bp	NC_013617	[8]
*Muraenesox bagio*	Muraenesocidae	18,247 bp	NC_013614	[8]
*Kaupichthys hyoproroides*	Chlopsidae	16,662 bp	NC_013607	[8]
*Robinsia catherinae*	Chlopsidae	16,627 bp	NC_013633	[8]
*Nessorhamphus ingolfianus*	Derichthyidae	17,782 bp	NC_013608	[8]
*Coloconger cadenati*	Colocongridae	17,755 bp	NC_013606	[8]
*Moringua microchir **	Moringuidae	15,858 bp	NC_013602	[8]
*Moringua edwardsi*	Moringuidae	16,841 bp	NC_013622	[8]
*Ophichthus rotundus*	Ophichthidae	17,785 bp	KY081397	[5]
*Pythonichthys microphthalmus*	Heterenchelyidae	17,042 bp	NC_013601	[8]
*Serrivomer sector*	Serrivomeridae	16,099 bp	NC_013436	[8]
*Stemonidium hypomelas*	Serrivomeridae	16,566 bp	NC_013628	[8]
*Anguilla marmorata*	Anguillidae	16,714 bp	NC_006540	[41]
*Anguilla interioris*	Anguillidae	16,713 bp	NC_006539	[41]
*Anguilla dieffenbachii*	Anguillidae	16,687 bp	NC_006538	[41]
*Simenchelys parasitica*	Synaphobranchidae	16,689 bp	NC_013605	[4]
*Synaphobranchus kaupii*	Synaphobranchidae	16,166 bp	NC_005805	[42]
*Ilyophis brunneus*	Synaphobranchidae	16,682 bp	NC_013634	[4]
*Myroconger compressus*	Myrocongridae	16,642 bp	NC_013631	[4]
*Scuticaria tigrina*	Muraenidae	16,521 bp	KP874183	[6]
*Gymnothorax minor*	Muraenidae	16,575 bp	MK204283	[42]
*Rhinomuraena quaesita*	Muraenidae	16,566 bp	NC_013610	[6]
*Eurypharynx pelecanoides*	Eurypharyngidae	18,978 bp	AB046473	[43]
*Saccopharynx lavenbergi*	Saccopharyngidae	18,495 bp	AB047825	[43]

Note: *** the length excludes the control region.

**Table 2 biology-12-00348-t002:** Features of the mitochondrial genomes of *A. meeki* and *A. shiroanago*.

Mitogenome	Position from/to		Length (bp)	Amino Acid	Start/Stop Codon	Anticodon	Intergenic Region from to (bp) *	Strand #
** *A. meeki* **								
*tRNA-Phe (F)*	1	71	71			GAA	0	H
*12S RNA*	72	1024	953				0	H
*tRNA-Val (V)*	1025	1095	71			TAC	0	H
*16S RNA*	1096	2794	1699				0	H
*tRNA-Leu^UUA^ (L1)*	2795	2870	76			TAA	0	H
*ND1*	2871	3839	969	323	ATG/TAG		0	H
*tRNA-Ile (I)*	3839	3906	68			GAT	−1	H
*tRNA-Gln (Q)*	3908	3978	71			TTG	1	L
*tRNA-Met (M)*	3978	4046	69			CAT	−1	H
*ND2*	4047	5093	1047	349	ATG/TAG		0	H
*tRNA-Trp (W)*	5092	5163	72			TCA	−2	H
*tRNA-Ala (A)*	5165	5233	69			TGC	1	L
*tRNA-Asn (N)*	5235	5307	73			GTT	1	L
*tRNA-Cys (C)*	5347	5410	64			GCA	39	L
*tRNA-Tyr (Y)*	5411	5481	71			GTA	0	L
*COX1*	5483	7090	1608	536	GTG/AGA		1	H
*tRNA-Ser^UCA^ (S1)*	7086	7156	71			TGA	−5	L
*tRNA-Asp (D)*	7162	7231	70			GTC	5	L
*COX2*	7237	7927	691	230	ATG/T		5	H
*tRNA-Lys (K)*	7928	8002	75			TTT	0	H
*ATP8*	8004	8171	168	56	ATG/TAA		1	H
*ATP6*	8162	8845	684	228	ATG/TAA		−10	H
*COX3*	8845	9630	786	262	ATG/TAA		−1	H
*tRNA-Gly (G)*	9630	9701	72			TCC	−1	H
*ND3*	9702	10,052	351	117	ATG/TAG		0	H
*tRNA-Arg (R)*	10,051	10,120	70			TCG	0	H
*ND4L*	10,121	10,417	297	99	ATG/TAA		0	H
*ND4*	10,411	11,791	1381	460	ATG/T		−7	H
*tRNA-His (H)*	11,792	11,862	71			GTG	0	H
*tRNA-Ser^AGC^ (S2)*	11,863	11,933	71			GCT	0	H
*tRNA-Leu^CUA^ (L2)*	11,934	12,007	74			TAG	0	H
*ND5*	12,008	13,846	1839	613	ATG/TAG		0	H
*Cyt b*	13,896	15,035	1140	380	ATG/TAA		49	H
*tRNA-Thr* *(T)*	15,037	15,107	71			TGT	1	H
D-loop 1	15,108	16,076	969				0	H
*ND6*	16,077	16,592	516	172	ATG/TAG		0	L
*tRNA-Glu (E)*	16,593	16,661	69			TTC	0	L
*tRNA-Pro (P)*	16,665	16,735	71			TGG	3	L
D-loop 2	16,736	17,695	960				0	H
** *A. shiroanago* **								
*tRNA-Phe (F)*	1	70	70			GAA	0	H
*12S RNA*	71	1032	962				0	H
*tRNA-Val (V)*	1033	1103	71			TAC	0	H
*16S RNA*	1104	2828	1725				0	H
*tRNA-Leu^UUA^ (L1)*	2823	2899	77			TAA	−6	H
*ND1*	2900	3862	963	321	ATG/TAG		0	H
*tRNA-Ile (I)*	3862	3929	68			GAT	−1	H
*tRNA-Gln (Q)*	3931	4001	71			TTG	1	L
*tRNA-Met (M)*	4001	4046	46			CAT	−1	H
*ND2*	4070	5116	1047	349	ATG/TAG		23	H
*tRNA-Trp (W)*	5115	5187	73			TCA	−2	H
*tRNA-Ala (A)*	5189	5257	69			TGC	1	L
*tRNA-Asn (N)*	5259	5331	73			GTT	1	L
*tRNA-Cys (C)*	5365	5430	66			GCA	33	L
*tRNA-Tyr (Y)*	5431	5501	71			GTA	0	L
*COX1*	5503	7110	1608	536	GTG/AGG		1	H
*tRNA-Ser^UCA^ (S1)*	7106	7176	71			TGA	−5	L
*tRNA-Asp (D)*	7182	7253	72			GTC	5	H
*COX2*	7258	7948	691	230	ATG/T		4	H
*tRNA-Lys (K)*	7949	8023	75			TTT	0	H
*ATP8*	8025	8192	168	56	ATG/TAA		1	H
*ATP6*	8183	8866	684	228	ATG/TAA		−10	H
*COX3*	8866	9651	786	262	ATG/TAA		−1	H
*tRNA-Gly (G)*	9651	9722	72			TCC	−1	H
*ND3*	9723	10,073	351	117	ATG/TAG		0	H
*tRNA-Arg (R)*	10,072	10,141	70			TCG	−2	H
*ND4L*	10,142	10,438	297	99	ATG/TAA		0	H
*ND4*	10,432	11,811	1380	460	ATG/TAA		−7	H
*tRNA-His (H)*	11,813	11,881	69			GTG	1	H
*tRNA-Ser^AGC^ (S2)*	11,882	11,952	71			GCT	0	H
*tRNA-Leu^CUA^ (L2)*	11,953	12,026	74			TAG	0	H
*ND5*	12,027	13,877	1851	617	ATG/TAA		0	H
*Cyt b*	13,919	15,058	1140	380	ATG/TAA		41	H
*tRNA-Thr (T)*	15,061	15,132	72			TGT	2	H
*ND6*	15,287	15,802	516	172	ATG/TAG		154	L
*tRNA-Glu (E)*	15,803	15,871	69			TTC	0	L
*tRNA-Pro (P)*	15,886	15,955	70			TGG	14	L
D-loop	15,956	16,922	967				0	H

* Intergenic region: non-coding bases between the feature on the same line and the line below, with a negative number indicating an overlap. ^#^ H: heavy strand; L: light strand.

**Table 3 biology-12-00348-t003:** Composition and skewness of the *A. meeki* and *A. shiroanago* mitogenomes.

	T	C	A	G	A + T%	AT-Skew	GC-Skew	Length (bp)
** *A. meeki* **								
Mitogenome	26.14	25.39	28.93	19.54	55.07	0.05	−0.13	17,695
*ND1*	28.48	28.90	20.95	21.67	49.43	−0.15	−0.14	969
*ND2*	28.27	25.21	26.27	20.25	54.54	−0.04	−0.11	1047
*COX1*	29.42	24.13	27.42	19.03	56.84	−0.04	−0.12	1608
*COX2*	27.49	23.60	30.39	18.52	57.88	0.05	−0.12	691
*ATP8*	26.79	25.00	34.52	13.69	61.31	0.13	−0.29	168
*ATP6*	28.80	28.51	28.65	14.04	57.45	0.00	−0.34	684
*COX3*	28.50	27.36	24.04	20.10	52.54	−0.09	−0.15	786
*ND3*	29.34	29.35	22.22	19.09	51.56	−0.14	−0.21	351
*ND4*	28.24	25.13	27.73	18.90	55.97	−0.01	−0.14	1381
*ND4L*	26.94	30.30	26.60	16.16	53.54	−0.01	−0.30	297
*ND5*	26.97	28.11	25.72	19.12	52.69	−0.02	−0.19	1839
*Cytb*	29.47	27.81	23.95	18.77	53.42	−0.10	−0.19	1140
*ND6*	33.33	20.93	14.53	31.21	47.86	−0.39	0.20	516
*tRNA*	27.12	20.45	29.03	23.40	56.15	0.03	0.07	1560
*rRNA*	20.06	22.25	33.78	23.91	53.84	0.26	0.04	2652
D-loop	25.82	22.29	36.18	15.71	62.00	0.17	−0.17	1929
** *A. shiroanago* **								
Mitogenome	26.85	23.90	32.54	16.71	59.38	0.10	−0.18	16,922
*ND1*	29.08	24.92	28.87	17.13	57.94	0.00	−0.19	963
*ND2*	26.46	24.45	33.72	15.38	60.17	0.12	−0.23	1047
*COX1*	31.59	22.26	27.80	18.35	59.39	−0.06	−0.10	1608
*COX2*	28.51	23.30	31.26	16.93	59.77	0.05	−0.16	691
*ATP8*	26.79	26.19	38.69	8.33	65.48	0.18	−0.52	168
*ATP6*	30.56	26.32	29.97	13.16	60.53	−0.01	−0.33	684
*COX3*	28.75	24.94	28.24	18.07	57.00	−0.01	−0.16	786
*ND3*	31.91	25.07	27.92	15.10	59.83	−0.07	−0.25	351
*ND4*	29.13	25.94	29.57	15.36	58.70	0.01	−0.26	1380
*ND4L*	32.32	25.59	27.61	14.48	59.93	−0.08	−0.28	297
*ND5*	28.31	25.07	32.36	14.26	60.67	0.07	−0.27	1851
*Cytb*	30.44	24.82	29.65	15.09	60.09	−0.01	−0.24	1140
*ND6*	41.67	14.73	14.92	28.68	56.59	−0.47	0.32	516
*tRNA*	28.17	20.23	29.71	21.90	57.87	0.03	0.04	1562
*rRNA*	20.54	22.29	36.55	20.62	57.09	0.28	−0.04	2687
D-loop	26.37	17.89	40.43	15.31	66.80	0.21	−0.08	967

## Data Availability

Due to ethical restrictions, the data presented in this study are available to researchers eligible under the Research Ethics Board rules on request from the corresponding author.

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
