# Peer review of "Comparative Analysis of Complete Mitochondrial Genome of Ariosoma meeki (Jordan and Snider, 1900), Revealing Gene Rearrangement and the Phylogenetic Relationships of Anguilliformes"

_biology, 2023, doi:10.3390/biology12030348_

Round 1

Author Response

Dear Reviewer:

Thank you very much for your review and suggestions! We answered your question with gratitude. We believe that after correction, the ideas discussed in this paper could be expressed more logically.

The following is our reply to your valuable suggestions:

Q1:  “According to the rules of taxonomy, every time a species is cited, its name must be written out in full and must accompany the author who described the species. In this case, the title should look like this: Comparative analysis of complete mitochondrial genome of Ariosoma meeki (Jordan & Snider, 1900) revealed gene rearrangement and phylogenetic relationships of Anguilliformes.”

A1:  Thank you very much. After our discussion, we have made the corrections according to your kind suggestions.

Q2:  Modify “two suborders” by “2 suborders”.

A2:  Thank you very much. It has been modified now.

Q3:  “I think it would be interesting to put a picture that corresponds to the traditional gene arrangement and special gene rearrangement in Anguiliformes, so that the patterns described for the group are clear.”

A3:  Thank you very much. Base on your suggestions, the picture (Fig.7) has been changed for a clear presentation

Q4:  Remove “in 1998”.

A4:  Thank you very much. It has been removed now.

Q5:  “In the last paragraph remove the "(accession number: OK585090)", this information is not appropriate here. Put the information in the material and methods or results.”

A5:  Thank you very much. Modifications have been made according to your kind suggestions.

Q6:  “In the last paragrafh “In this paper, we reported a new version of the complete mitochondrial genome sequence of A. meeki (accession number : OK585090), the genetic composition and arrangement of A. meeki were described in detail and the comparative genomic analysis was conducted among A. meeki (this study) and A. shiroanago” I would change the text to “In this paper, we report a new version of the complete mitochondrial genome sequence of Ariosoma meeki, and analyzed the genetic composition and arrangement described for the species, and performed a comparative genomic analysis between A. meeki (this study) and Ariosoma shiroanago.”. Remember that every time a new species is mentioned in the text, its name must be written out in full, regardless of whether it is the same genus as another species already presented.”

A6:  Thank you very much for your valuable suggestions, they will undoubtedly improve the readability of this paper. Modifications have been made.

Q7:  “This part “Interestingly, gene rearrangement was detected in both Ariosoma mitogenomes, differed from the genetic features of two published A. meeki mitogenomes (KX641476 and MN616974) which have been marked as “UNVERIFIED” by NCBI. The arrangement of the mitochondrial genome is consistent in most vertebrate mitogenomes, while two kinds of gene arrangements have been revealed in the order Anguilliformes, which leads us to some confusion about the involved mechanism. In particular, some Anguilliformes population was known to converge into nonmonophyletic results such as species in Congridae and Nettastomatidae [29]. As of now, there is no comprehensive elaboration to this.” should be part of its results and discussion, and should not be presented in the introduction and the part “The relevant research is in its infancy [30]. In light of the above background and issues, there is an urgent need for a systematic analysis of Anguilliformes.” It should be in conclusions.”

A7:  Thank you very much. Looking back on the whole paper, we believe that the quality of this paper will reach into a new level after adopting your valuable suggestions. Therefore, we have made logical modifications to the whole paper according to your suggestions.

Q8:  “In the topic 2.2, it is important to put the number of samples.”

A8:  Thank you very much, we added them according to your suggestion.

Q9:  “In the sentence “The phylogenetic analyses were conducted utilizing PhyML80and MrBayes 3.2.6 software based on Bayesian inference (BI) and maximum likelihood (ML) methods [45][46].” Change by “The phylogenetic analyses were conducted utilizing MrBayes 3.2.6 and PhyML80 software based on Bayesian inference (BI) and maximum likelihood (ML), respectively[45][46].”

A9:  Thank you very much, modifications have been made.

Q10:  “How did you define the rearrangements in the A. meeki genome? Why didn't you use specific tools, for example CREx software (https://doi.org/10.1093/bioinformatics/btm468), to define the evolutionary events that generated the gene rearrangements?”

A10:  Thank you for your reminder. We defined the rearrangements using CREx software, and found that A. meeki mitogenome experienced TDRL scenario. Now we add this new statement in the text of Materials and Methods.

Q11:  In the topic 3.1, change “one origin of replication” by “1 origin of replication”.

A11:  Thank you very much. Modifications have been made.

Q12:  “A very particular suggestion, I don't see the need for the use of figures 3 and 4. They are just taking up space, I suggest removing them.”

A12:  Thank you very much for your advice. However, we believe that, on the one hand, the traditional method is to find out and verify the location of tRNA in mitochondrial DNA first, and many online programs or software will perform this function. In this study, we also used this method. On the other hand, as an important component of mitochondrial DNA, tRNA is as important as PCGs. We believe that correctly and clearly showing the structure comparison of tRNAs can not only express the relationship between Ariosoma shironango and Ariosoma meeki, but also serve as evidence for the sequence rearrangement of species of the same genus. The sequence results show that (Fig. 8) (Fig. 9), Ariosoma shiroanago is probably also a rearranged species. We cannot directly prove that Ariosoma shiroanago is a rearranged species, because we have not collected samples of Ariosoma shiroanago. We can't bear enough responsibility to make this conclusion, so we hope to tell the reader that this is a possible result through the published comparison of correct mitochondrial DNA fragments. Therefore, Fig 3 and Fig 4 as with other evidence, as part of a complete comparison. This is the reason why we keep these two pictures. I sincerely hope you can accept our explanation.

Q13:  “In this sentence “In the mitogenome of A. meeki, an additional control region was detected, and two coding proteins Cytb and ND6 underwent translocation. Compared with other nonrearranged species, such a genetic phenomenon between Cytb and ND6 can be found over the published Ariosoma mitogenome (GenBank accession number: AP010861).” I don't know if it was an error in my interpretation, but there is no translocation of Cytb, only of the ND6 gene and other non-PCGs.”

A13:  Thank you very much. This is an an error in our statement, and we have made the correct modifications.

Q13:  “In the continuation of the sentence I quote above, it says Ariosoma shiroanago, change it to A. shiroanago.”

A13:  Thank you very much. Modifications have been made.

Q14:  “Like the number of species in the analysis is low, I would change the style of the tree to a vertical tree rather than a circular tree. I think it would be more appropriate for the reader to visualize the relationships.”

A14:  Thank you very much for your advice. To be honest, in our first edition article, it is indeed a traditional vertical tree. However, when consulting the literature, we found that the round tree can help readers quickly understand the information that the evolutionary tree wants to express. Compared with traditional trees, round trees can more intuitively reflect the clustering differences between genetically rearranged and non genetically rearranged species. In our tree, we use different colors to express the clustering of species closer to the trunk position. We hope this is a faster and more convenient way to reflect the results of the evolutionary tree. At the same time, compared with traditional trees, round trees can show more information and are more beautiful. I sincerely hope you can accept our explanation.

Q15:  “I get confused with the phylogenetic tree results, see, you say that you performed Bayesian inference (BI) and maximum likelihood (ML) analyses. However, the tree you present has only the posterior probability values (ranging from 0 to 1), but you call this the bootstrap value, which would correspond to the maximum likelihood values (ranging from 0 to 100%). Therefore, I believe that there are errors in the adoption of these values in the tree, that being said, I ask that both the probability a posteriori (BI) and bootstrap (ML) values be presented in the updated image of the tree.”

A15:  Thank you very much for your comments. The phylogenetic tree provided previously was an initial version and lack of ML parameters, now it has been replaced by a new tree(Fig.8).

Reviewer 2 Report

Huang and colleagues have sequenced the complete mitogenome of Ariosoma meeki and discussed the gene rearrangement in Anguilliformes. I don’t think the manuscript is acceptable for publication in its current version.

Major concerns:

1. Similar gene rearrangement in mitochondrial genomes of Anguilliformes has already been discussed (such as Reference 31). In addition, this study and Ref 31 almost have the same organization, and quite similar results (including the tables and figures), thus the novelty of this research is debatable.

2. Although the manuscript focuses on “Comparative analysis of complete mitochondrial genome”, it just compares two mitogenomes (A. meeki and A. shiroanago). According to previous researches (e.g. mentioned above), and the long mitogenome size of some families (e.g. Congridae, Muraenesocidae, Nettastomatidae) in Table 1, it is possible that a similar change may have happened in all these families. In addition, combined with the phylogenetic tree, it would also be possible to trace in what clade (including the ancestor), and when (and how many times if they evolved independently) this change happened. Such results would be more interesting for reading.

3. Page1. Par1. “Approximately 4% of the reported fish… , involving 15 subjects and 34 families[4], according to … NCBI database”. I’m interested in these data but the cited reference [4] does not provide the info. Please check and cite the correct one. In addition, what does the “15 subjects” mean?

4. I have no idea why two A. meeki mitogenomes have such big difference. But to me, there probably is an assembly or annotation error in sample KX641476, as suggested by the authors in Page 14 : “The variation in mitogenome length may be attributed to incomplete sequencing or annotation”. In addition, I agree with the authors that the missing of extra D-loop in A. shiroanago should also be an error. Please check more mitogenome sequences and re-annotate them to see if the guess is right.

5. Regarding the rearrangements, many hypotheses have been proposed, such as Ref 31 and 51. Please add more discussions about the different hypotheses and provide more evidences for your TDRL model (in addition to the “a considerable interval of 49bp next to the rearranged Cytb”).

6. There are numerous grammatical errors in the manuscript. Language editing is highly recommended. Please also add the line numbers in the manuscript for ease of reviewing.

Minor concerns:

1. Why Figure 1 was placed between Figure 3 and 4?

2. Figure 2. should be amino acid composition of the proteins. What does the Y-axis refer to? and what about the colored bars in C and D?

3. Why don’t you align the sequences?

4. Figure 6. “13 thirteen”

5. Grammar errors such as in Abstract: a. (no subject) not only possessed two control regions but genes… (which is a full sentence.) ; b. This study was expected to provide

Author Response

Dear Reviewer:

Thank you very much for your comments which inspired us a lot! We sincerely hope that this study will be given more thought after providing our explanation.

The following is our reply to these valuable suggestions:

Q1:  “Similar gene rearrangement in mitochondrial genomes of Anguilliformes has already been discussed (such as Reference 31). In addition, this study and Ref 31 almost have the same organization, and quite similar results (including the tables and figures), thus the novelty of this research is debatable.”

A1:  Thank you very much for your comments! For gene rearrangement in mitochondrial genes of Anguilliformes has already been discussed, this is the analysis process piled up by quantity. For the process of bioinformatics to the evolution of species, the gene sequence of each species is particularly important. In Anguilliformes, the mitochondrial genome disclosed so far is still relatively few for other subjects. On the one hand, it is difficult to obtain species samples of Anguilliformes; on the other hand, the mitochondrial genome of some species in Anguilliformes is different from the traditional genome structure. The evolution and evolution of species can only be truly calculated by combining geological changes, and it can also be mutually verified with geological information. The target species in this paper is A meeki, which has been marked as an error sequence on NCBI. Unfortunately, gene rearrangement has never been mentioned in A meeki. The main purpose of this study is to publish the correct mitochondrial genome sequence and correct  the position of A. meeki in the gene rearrangement of Anguilliformes. We think the analysis of tables and charts are necessary, as the results of these analyses can support the position of organisms in the evolutionary tree and also verify the kinship between species. At the same time, we modified the evolutionary tree for a better understanding to the difference between two types of gene arrangement population in Anguilliformes. 

Q2:  “Although the manuscript focuses on “Comparative analysis of complete mitochondrial genome”, it just compares two mitogenomes (A. meeki and A. shiroanago). According to previous researches (e.g. mentioned above), and the long mitogenome size of some families (e.g. Congridae, Muraenesocidae, Nettastomatidae) in Table 1, it is possible that a similar change may have happened in all these families. In addition, combined with the phylogenetic tree, it would also be possible to trace in what clade (including the ancestor), and when (and how many times if they evolved independently) this change happened. Such results would be more interesting for reading.”

A2:  Thank you very much for your question! First of all, the focus of this study, as you said, is the comparative analysis of complete mitochondrial genomes. Since the two published A meeki mitogenomes in NCBI have been marked as “UNVERIFIED”, we carried out the comparison between A. meeki and another Ariosoma mitogenome. After the comparative analysis between them, we infered that Ariosoma species (eg: A meeki and A. shiroanago) experenced gene rearrangement events, and A. shiroanago mitogenome in NCBI may lack a part of control regions. It is possible that a similar change may have happened in other families. Thus, we modified the Fig.7, it shows that the same gene rearrangement phenomenon occurs in other families (eg: Nettastomatidae, Muraenesocidae,  Colocongridae). Combined with the modified phylogenetic tree (Fig.8), we found that these families with the gene rearrangement formed a independent clade, and the remaining families (eg: Anguillidae, Synaphobranchidae, Muraenidae) with typical vertebrate gene arrangement formed another clade. We think that this result implied the evolution and origin of the diversified Anguilliformes species as this pattern of presence/absence of gene arrangement may be a potential phylogenetic marker for identification of this morphologically conservative group of fish at the suborder-level

Q3:  “Page1. Par1. “Approximately 4% of the reported fish… , involving 15 subjects and 34 families[4], according to … NCBI database”. I’m interested in these data but the cited reference [4] does not provide the info. Please check and cite the correct one. In addition, what does the “15 subjects” mean?”

A3:  I'm sorry about that , we corrected it by a new reference. “15 subjects“ was a wrong presentation, we removed it.

Q4:  “I have no idea why two A. meeki mitogenomes have such big difference. But to me, there probably is an assembly or annotation error in sample KX641476, as suggested by the authors in Page 14 : “The variation in mitogenome length may be attributed to incomplete sequencing or annotation”. In addition, I agree with the authors that the missing of extra D-loop in A. shiroanago should also be an error. Please check more mitogenome sequences and re-annotate them to see if the guess is right.”

A4:  Thank you very much for your kind suggestion. As the A. meeki belongs to the family Congridae, we checked other Congridae mitogenome sequences such as Paraconger notialis (NC_013630), Heteroconger hassi (NC_013629), and Conger japonicu (KR131863). After re-annotation of these Congridae mitogenomes, we found that all of them reflect the same gene order as our A. meeki mitogenome, and two Dloops were detected. In this case, we guess that the variation in published Ariosoma mitogenome length may be attributed to incomplete sequencing or annotation. However, these are just conjectures. Considering there are few available Ariosoma mitogenomes, we dare not make such definite conclusion based on limited data. After all, this is a very rigorous and sensitive scientific issue.

Q5:  “Regarding the rearrangements, many hypotheses have been proposed, such as Ref 31 and 51. Please add more discussions about the different hypotheses and provide more evidences for your TDRL model (in addition to the “a considerable interval of 49bp next to the rearranged Cytb”).”

A5:  Thank you very much for your kind suggestion.

The model based on the recombination hypothesis was initially proposed for gene rearrangement in the nuclear genome and generally adopted to explain small fragment exchanges and inversions events in mitogenomes. However, our comparative analysis shows that the Ariosoma mitogenomes have undergone genes translocation and genome-scale expansion Therefore, the discovery of mitochondrial gene rearrangement in Ariosoma mitogenomes is far-fetched to be explained by this model.

The TDNL model attached emphasis on the non-random loss. In this case, the duplication followed by the loss of genes is predesigned resort to corresponding transcriptional polarity and position in the genome, genes are clustered in the same polarity (Light or Heavy strand coding) and the gene order remain unchanged (for example, the GCT cannot be produced by gene loss alone during the duplication from TCG to TCGTCG) . This hypothesis does not apply to our findings, as the structural variation in Ariosoma mitogenomes is not the result of alterations in the genes' transcriptional polarity.

The TDRL model underlined the rearrangement based on the incomplete deletion of repeated genes. In A. meeki mitogenome, an extra Dloop was found in the downstream of tRNA-Thr, both Dloops revealed a high degree of resemblance in nucleotide composition and base arrangement. In addition, ND6 combined with tRNA-Glu genes were translocated from the stream of downstream of ND5 onto the upstream of tRNA-Pro. In the TDRL model, the intergenic spacers or pseudogenes were commonly detected in the rearrangement region. In the mitochondrial genome of A. meeki, the existence of a considerable interval among ND5 and Cytb further indicated the bias to this model (Table 2). The TDRL model underlined the rearrangement based on the incomplete deletion of repeated genes. In A. meeki mitogenome, an extra Dloop was found in the downstream of tRNA-Thr, both Dloops revealed a high degree of resemblance in nucleotide composition and base arrangement. In addition, ND6 combined with tRNA-Glu genes were translocated from the stream of downstream of ND5 onto the upstream of tRNA-Pro. In the TDRL model, the intergenic spacers or pseudogenes were commonly detected in the rearrangement region. In the mitochondrial genome of A. meeki, the existence of a considerable interval among ND5 and Cytb further indicated the bias to this model (Table 2).

(The above modified content is now added to the text)

Q6:  “There are numerous grammatical errors in the manuscript. Language editing is highly recommended. Please also add the line numbers in the manuscript for ease of reviewing.”

A6:  Thank you very much for your kind suggestion. We tried our best and made some improvements again. If it still needs to be modified later, we will take the initiative to contact the editor. With regard to the line numbers (It may be a misunderstanding caused by the software version), we have marked the line number in the uploaded Word version file, and we will upload a PDF version of the document again, hoping to explain this misunderstanding.

Minor concerns:

Q1:  Why Figure 1 was placed between Figure 3 and 4?

A1:  Thank you very much. This may be an error caused by the different software version, I think this error could be removed in the new PDF file.

Q2:  Figure 2. should be amino acid composition of the proteins. What does the Y-axis refer to? and what about the colored bars in C and D?

A2:  Thank you very much. Y-axis refers to the number of each amino acid in 13 PCGs. The colored bars in C and D represent the relative synonymous codon usage (RSCU) analysis. The height of each color column represents the usage frequency of corresponding amino acid codons in 13 PCGs. Different colors represent different codons in amino acids.  

(It was our mistake, we have revised the caption and hope it can be more clear.)

Q3:  Why don’t you align the sequences?

A3:  This may be a misunderstanding, and the new PDF file can explain this error.

Q4:  Figure 6. “13 thirteen”

A4:  Thank you very much. We have modified it.

Q5:  Grammar errors such as in Abstract: a. (no subject) not only possessed two control regions but genes… (which is a full sentence.) ; b. This study was expected to provide

A5:  Thank you very much for your careful review. They were our grammatical mistakes, we revised them.

Reviewer 3 Report

The authors reported and analyzed the complete mitochondrial genome of A. meeki,comparing it with the published mitogenomes of A. meeki and congeneric species, and demonstrated the phylogenetic relationship among Anguilliformes species.This study is piece of nice work and the manuscript is well written.  However, the manuscript should be improved before it is accepted as a publication in the Journal, Biology MDPI.

The major:

The authors should demonstrate the biological significance of the rearrangement of mitochondial genome during teleostean species evolution.

Author Response

Thank you very much for your kind suggestion. We tried our best and made some improvements again. If it still needs to be modified later, we will take the initiative to contact the editor.

Since the two published A meeki mitogenomes in NCBI have been marked as “UNVERIFIED”, we carried out the comparison between A. meeki and another Ariosoma mitogenome. After the comparative analysis between them, we infered that Ariosoma species (eg: A meeki and A. shiroanago) experenced gene rearrangement events, and A. shiroanago mitogenome in NCBI may lack a part of control regions. It is possible that a similar change may have happened in other families. Thus, we modified the Fig.7, it shows that the same gene rearrangement phenomenon occurs in other families (eg: Nettastomatidae, Muraenesocidae, Colocongridae). Combined with the modified phylogenetic tree (Fig.8), we found that these families with the gene rearrangement formed a independent clade, and the remaining families (eg: Anguillidae, Synaphobranchidae, Muraenidae) with typical vertebrate gene arrangement formed another clade. We think that this result implied the evolution and origin of the diversified Anguilliformes species as this pattern of presence absence of gene arrangement may be a potential phylogenetic marker for identification of this morphologically conservative group of fish at the suborder-level.

Regarding the rearrangements, many hypotheses have been proposed:

The model based on the recombination hypothesis was initially proposed for gene rearrangement in the nuclear genome and generally adopted to explain small fragment exchanges and inversions events in mitogenomes. However, our comparative analysis shows that the Ariosoma mitogenomes have undergone genes translocation and genome-scale expansion Therefore, the discovery of mitochondrial gene rearrangement in Ariosoma mitogenomes is far-fetched to be explained by this model.

The TDNL model attached emphasis on the non-random loss. In this case, the duplication followed by the loss of genes is predesigned resort to corresponding transcriptional polarity and position in the genome, genes are clustered in the same polarity (Light or Heavy strand coding) and the gene order remain unchanged (for example, the GCT cannot be produced by gene loss alone during the duplication from TCG to TCGTCG). This hypothesis does not apply to our findings, as the structural variation in Ariosoma mitogenomes is not the result of alterations in the genes' transcriptional polarity.

The TDRL model underlined the rearrangement based on the incomplete deletion of repeated genes. In A. meeki mitogenome, an extra Dloop was found in the downstream of tRNA-Thr, both Dloops revealed a high degree of resemblance in nucleotide composition and base arrangement. In addition, ND6 combined with tRNA-Glu genes were translocated from the stream of downstream of ND5 onto the upstream of tRNA-Pro. In the TDRL model, the intergenic spacers or pseudogenes were commonly detected in the rearrangement region. In the mitochondrial genome of A. meeki, the existence of a considerable interval among ND5 and Cytb further indicated the bias to this model (Table 2). The TDRL model underlined the rearrangement based on the incomplete deletion of repeated genes [56]. In A. meeki mitogenome, an extra Dloop was found in the downstream of tRNA-Thr, both Dloops revealed a high degree of resemblance in nucleotide composition and base arrangement. In addition, ND6 combined with tRNA-Glu genes were translocated from the stream of downstream of ND5 onto the upstream of tRNA-Pro. In the TDRL model, the intergenic spacers or pseudogenes were commonly detected in the rearrangement region. In the mitochondrial genome of A. meeki, the existence of a considerable interval among ND5 and Cytb further indicated the bias to this model (Table 2). (The above modified content is now added to the text)

We think these results provide insight into gene arrangement features of Anguilliform mitogenomes and lay the foundation for further phylogenetic studies on Anguilliformes

Round 2

Reviewer 2 Report

The authors have addressed my concerns and I’m happy with most of the responses. But I still have a few concerns.

1. I totally understand that this manuscript is determined to provide a “correct” complete mitochondrial genome of A meeki, but if it takes most pages to show only the descriptive results (as presented from page 4-10) without any inferential statistics/conclusions, it might not be appropriate for the journal Biology. Therefore, it is highly recommended that the authors move some tables (Table 2, 3) and figures (Figure 3, 4, 6) into supplementary files and emphasize the differences and novel findings obtained from the “comparative” analysis.

2. Figure 5. Align the two sequences (using MEGA etc.) and present them in an alignment format (using Texshade etc.).

3. Figure 8. could be changed to a traditional tree and be merged to Figure 7. Put the tree on the left and the (modified) gene synteny on the right. Such a figure could clearly show how genes rearranged across the Anguilliformes tree.

4. This is not mandatory but could be useful. Download the original complete mitochondrial genomes of the studied Anguilliformes species, perform synteny analysis using DNA sequenced directly (no the annotated genes), and plot the result (using jcvi etc.).

5. What do you think about “a novel gene arrangement of L-strand coding genes” (Ref 41) and the TDRL mode you proposed. The two paths seem to result in almost the same (Dloop1-ND6-Dloop2 vs. CN2-ND6-CN1) gene rearrangements. A phylogeny of the D-loop regions may tell which one is the original sequence and which one is the new copy.

Author Response

Dear Reviewer:

Thank you so much for your comment! Please forgive my late reply, because the covid has caused irreversible damage to me and even normal scientific research activities can not be maintained. So this exchange is late, I am so sorry!

The following is our reply to these valuable suggestions:

Q1. I totally understand that this manuscript is determined to provide a “correct” complete mitochondrial genome of A meeki, but if it takes most pages to show only the descriptive results (as presented from page 4-10) without any inferential statistics/conclusions, it might not be appropriate for the journal Biology. Therefore, it is highly recommended that the authors move some tables (Table 2, 3) and figures (Figure 3, 4, 6) into supplementary files and emphasize the differences and novel findings obtained from the “comparative” analysis.

A1: Thank you very much for your comments! I fully understand the value of your comments and have thought deeply about it. After our discussion, we believe that a new sequence publication contains a description of various basic genetic information of this species. In general, after the traditional mitochondrial sequence is published, it will be analyzed simply by biological means, such as amino acid composition or two-dimensional structure of tRNA. Therefore, we take (Table 2, 3) (Figure 3, 4, 6) as an indispensable part of the article. At the same time, in order to highlight the universality of this study, we focus on A The mitochondrial rearrangement of meeki is also based on the similarity between other fragments and other species. The conservation of mitochondria is an important feature of maternal inheritance, but gene rearrangement gives more possibilities for analysis. We believe that the analysis of other fragments in addition to rearrangement is complementary to the specificity of gene rearrangement. Sorry, we still want to keep the status quo. Of course, if we need a shorter and more concise conclusion, we can also make some modifications.

Q2. Figure 5. Align the two sequences (using MEGA etc.) and present them in an alignment format (using Texshade etc.).

A2: Thank you very much for your comments! We have modified this.

Q3. Figure 8. could be changed to a traditional tree and be merged to Figure 7. Put the tree on the left and the (modified) gene synteny on the right. Such a figure could clearly show how genes rearranged across the Anguilliformes tree.

A3: This is a very contrastive suggestion! But in fact, there are too many elements and information in Figure 7. If Figure 7 and Figure 8 are juxtaposed into a picture, the picture will be very blurred. At the same time, the main expression of Figure 7 is the rearrangement process of each gene and the details of the beat model during gene rearrangement. Figure 8 is the information about the trees of some Anguilliformes species. We want to analyze the basis and discussion of mitochondrial gene rearrangement through these two analyses, but each analysis is still independent. Therefore, Figure 7 and Figure 8 are not suitable for splicing into a picture. so sorry! But we can still try to use your suggestions in the next article. We have to admit that this is a very good proposal, but we want to keep it as it is.

Q4. This is not mandatory but could be useful. Download the original complete mitochondrial genomes of the studied Anguilliformes species, perform synteny analysis using DNA sequenced directly (no the annotated genes), and plot the result (using jcvi etc.).

A4: Thank you very much for your suggestion! We did a collinearity analysis, but the results did not meet expectations. The mitochondrial genome is approximately 17000 bp, and its gene composition is not repetitive. If we have done the collinearity analysis of the base sequence, the most appropriate is to analyze the deletion or addition of the target gene segment, which is not in line with your suggestion; If we do gene collinearity analysis, there is no duplication of multiple gene fragments in a complete mitochondrial genome; If we conduct collinearity analysis on different species, we are not very clear about the anchor point of analysis. After consulting the literature, we would like to ask you in detail where we should put the purpose anchor of collinearity analysis? Your suggestion provides us with a new direction, but there is no collinearity analysis of mitochondrial genome in previous years' literature, so we urgently hope that you can help us point out the right direction. Thank you again for your suggestion!

Q5. What do you think about “a novel gene arrangement of L-strand coding genes” (Ref 41) and the TDRL mode you proposed. The two paths seem to result in almost the same (Dloop1-ND6-Dloop2 vs. CN2-ND6-CN1) gene rearrangements. A phylogeny of the D-loop regions may tell which one is the original sequence and which one is the new copy.

A5: Thank you very much for your comments! This is indeed a problem worth studying. In this study, our proposed model is that Dloop2 is the original fragment Dloop1 is copied. We have not provided corresponding evidence, and previous research has not provided reasonable evidence. We built trees by single Dloop on the species in this study, but the results were not very ideal. I'm sorry we can't provide more accurate evidence. But your suggestion is worth studying. Therefore, we have carried out a next experimental plan to make a coincidence rate analysis of common habitat of species by studying the range of habitat of species. At the same time, compare and analyze the two sections of Dloop to find evidence of the order of Dloop2 and Dloop1. However, this is a very time-consuming research program, and it is not known whether the correct conclusions can be drawn. We communicated with the editor, and the follow-up research results will continue to be published. Thank you very much for your constructive comments. We are deeply inspired!
